# The Association of *ADAMTS7* Gene Polymorphisms with the Risk of Coronary Artery Disease Occurrence and Cardiovascular Survival in the Polish Population: A Case-Control and a Prospective Cohort Study

**DOI:** 10.3390/ijms25042274

**Published:** 2024-02-14

**Authors:** Joanna Iwanicka, Anna Balcerzyk-Matić, Tomasz Iwanicki, Katarzyna Mizia-Stec, Paweł Bańka, Artur Filipecki, Katarzyna Gawron, Alicja Jarosz, Tomasz Nowak, Jolanta Krauze, Paweł Niemiec

**Affiliations:** 1Department of Biochemistry and Medical Genetics, School of Health Sciences in Katowice, Medical University of Silesia, Medykow Street 18, 40-752 Katowice, Poland; abalcerzyk@sum.edu.pl (A.B.-M.); tiwanicki@sum.edu.pl (T.I.); alicja.jarosz@sum.edu.pl (A.J.); tnowak@sum.edu.pl (T.N.); pniemiec@sum.edu.pl (P.N.); 2First Department of Cardiology, School of Medicine in Katowice, Medical University of Silesia, 47 Ziołowa St., 40-635 Katowice, Poland; kmizia-stec@sum.edu.pl (K.M.-S.); pawelbanka19@gmail.com (P.B.); afilipecki@sum.edu.pl (A.F.); 3Department of Molecular Biology and Genetics, Faculty of Medical Sciences in Katowice, Medical University of Silesia, Medykow 18, 40-752 Katowice, Poland; kgawron@sum.edu.pl; 41st Department of Cardiac Surgery/2nd Department of Cardiology, American Heart of Poland, S. A. Armii Krajowej 101, 43-316 Bielsko-Biala, Poland; jolakra@poczta.fm

**Keywords:** coronary artery disease, polymorphism, *ADAMTS7*, lipids, mortality

## Abstract

The aim of this study was to investigate whether the polymorphisms of the *ADAMTS7* gene affect the risk of occurrence and mortality due to CAD. The study group included 231 patients diagnosed with CAD and 240 control blood donors. The genotyping of specified polymorphisms, i.e., rs1994016, rs3825807, and rs7173743, was performed using the TaqMan-PCR. We found that the C allele carriers of the rs1994016 and A allele carriers of the rs3825807 polymorphisms increased the risk of CAD, respectively: OR = 1.72, *p* = 0.036; OR = 1.64, *p* = 0.04. Moreover, we studied the biological interactions of specified variants, i.e., rs3825807, rs1994016, and rs7173743, and previously approved risk factors of CAD. We demonstrated here that selected polymorphisms of *ADAMTS7* increased the risk of CAD altogether with abnormalities of total cholesterol and LDL concentrations in serum. Although survival analyses did not reveal statistical significance, we observed a trend for the AA genotype of the rs3825807 *ADAMTS7*, which may predispose to death due to CAD in a 5-year follow-up. In conclusion, the *ADAMTS7* polymorphisms investigated in this study may increase the risk of occurrence and/or death due to CAD in the Polish population.

## 1. Introduction

Coronary artery disease (CAD) is one of the highest mortality diseases in the world. It has been estimated that by the end of 2030, about 12 million people will die from coronary atherosclerosis [1].

Therefore, there is still an urgent need to search for genetic factors and biomarkers that not only influence the manifestation of the disease but also predispose individuals to premature death due to cardiovascular episodes [2]. So far, 58 independent loci associated with CAD have been identified in genome-wide association studies (GWAS). Polymorphic variants of selected genes are mainly related to the inflammatory response, regulation of oxidative stress, lipid function, endothelial dysfunction, and other pathogenic processes involved in the progression of atherosclerotic lesions. Among novel biomarkers, hsCRP has emerged as the most promising in chronic situations; others need further clinical studies. Moreover, many environmental risk factors may result in epigenetic modifications with abnormal phenotypic expression of genetic information, resulting in an increased risk of cardiovascular diseases. Recent advances in epigenetics point to the importance of DNA methylation profiles, chemical modifications of histones, and the role of micro-RNAs as promising atherosclerosis indicators. Numerous clinical studies have also shown an association between exposure to various environmental factors during life or the intrauterine period and the increased occurrence of selected biomarkers of atherosclerosis. A better understanding of long-term environmental impacts should provide more effective medical interventions to reduce the incidence of cardiovascular disease [3]. Unlike serum and genetic biomarkers, ultrasound is better studied and already has a high degree of clinical application. Measurements of carotid complex-media thickness, carotid plaque detection, flow-dependent dilation, and arterial stiffness can be used to predict cardiovascular events and detect preclinical atherosclerosis. All such methods can also be easily and inexpensively obtained with a high degree of reproducibility [4].

*ADAMTS7* (15q24.2) belongs to the group of genes encoding metalloproteinases (MMPs) with proteolytic activity against extracellular substrates, primarily extracellular matrix (ECM) proteins. Specific single nucleotide polymorphisms (SNPs) of *ADAMTS7* may promote the migration of smooth muscle cells in blood vessels, leading to the progression of atherosclerotic changes [5]. Simultaneously, genome-wide association studies (GWAS) have demonstrated an association of polymorphic variants of this gene with the clinical phenotype of CAD, making it an attractive target for therapeutic interventions [6,7].

Among numerous polymorphisms of the *ADAMTS7*, several seem of special interest. Briefly, the rs3825807 may influence the functionality of the encoded ADAMTS7 protein by affecting its maturation [7,8,9]. As a result, long-term observation of patients with CAD has shown that the AA genotype of the rs3825807 SNP was a significant independent risk factor for death due to the cardiovascular episodes compared to the GG homozygotes [5]. Otherwise, the GG homozygosity may reduce the risk of atherosclerotic progression and CAD [9]. The genetic significance of the rs3825807 *ADAMTS7* variant was also confirmed in the pathogenesis of CAD by GWAS [10]. Moreover, the rs1994016 was associated with CAD previously diagnosed by angiography [6,11]. It was also demonstrated, that the rs1994016 variant may predispose to peripheral artery disease by affecting gene expression and increased mRNA levels of *ADAMTS7* [12]. Furthermore, the rs7173743 intergenic variant, located 40 kb upstream of the transcription start site of the *ADAMTS7,* seems as well to be a promising candidate risk factor, as previous GWAS studies suggested an association of this polymorphism with CAD [13] and showed its significant role as a risk factor of unstable atherosclerotic plaque occurrence [14].

Importantly, so far, the *ADAMTS7* gene polymorphisms have not been studied in the context of CAD in the Polish population; therefore, the primary aim of this study was to investigate the impact of the rs1994016, rs3825807, and rs7173743 variants of the *ADAMTS7* gene on the risk of CAD in the case–control model. The secondary aim was to evaluate the interactions between *ADAMTS7* polymorphisms, traditional risk factors, clinical phenotype of CAD, and the survival of patients over a 5- and 10-year perspective in a prospective cohort model.

## 2. Results

### 2.1. Analysis of the ADAMTS7 Polymorphisms

All analyzed genotype frequencies conformed to Hardy–Weinberg equilibrium (*p* > 0.05). The frequency of the C allele of the rs1994016 polymorphism was higher in the CAD group compared to the control group (OR = 1.72, 95% CI: 1.03–2.86, *p* = 0.036). Moreover, statistical significance was shown for TT homozygosity, which has a protective role against CAD (OR = 0.58, 95% CI: 0.35–0.97, *p* = 0.036). The power for 95% confidence intervals was 50.39%.

The frequencies of the A allele carriers of the rs3825807 polymorphism were higher in the CAD group in comparison to controls (OR = 1.67, 95% CI: 1.02–2.74, *p* = 0.042), and the GG homozygosity is associated with protection against CAD (OR = 0.60, 95% CI: 0.36–0.99, *p* = 0.042). The power for 95% confidence intervals was 52.74%.

In the case of the rs7173743 polymorphism, there were no statistically significant differences among allele and genotype frequencies of studied groups; however, we observed an increasing trend in the frequency of the T allele carriers in patients compared to controls (*p* = 0.068), which may predispose to CAD and indicate the potential protective role of CC homozygosity (*p* = 0.068). The genotype and allele frequencies of the *ADAMTS7* polymorphisms are shown in Table 1.

The haplotype analysis revealed the presence of an 8 kB diplotype block, which consists of rs1994016 and rs3825807 variants. The rs7173743 SNP was not found in linkage disequilibrium (LD) with any other SNPs (Figure 1). The CA and TG alleles (rs1994016 and rs3825807, in both cases) appeared in the strongest LD. The first diplotype was also characterized by the highest frequency in the study group (0.584%). Detailed data on the LD analysis and the frequency of haplotypes of the studied polymorphisms are presented in Figure 1. There was no association of the *ADAMTS7* gene haplotypes with CAD and its clinical phenotype.

### 2.2. The ADAMTS7 Polymorphisms Association with Clinical Phenotypes of CAD and Lipids Concentration

There were no statistically significant associations of *ADAMTS7* genotypic variants and history of myocardial infarction, advanced coronary atherosclerosis (presence of multivessel coronary disease or critical occlusion >90%) observed during coronary angiography, left ventricular hypertrophy, diabetes mellitus, or abnormal serum lipid concentration.

### 2.3. Interactions ADAMTS7 Polymorphisms with Previously Approved CAD Risk Factors

In the current section, the results of the analysis of gene-traditional risk factor interactions for CAD are presented, including interactions between genotypic variants of the *ADAMTS7* gene polymorphisms and hypertension, overweight/obesity, male gender, smoking, and lipid metabolism parameters. Only elevated levels of total cholesterol (TC) and LDL showed synergistic interactions with the polymorphisms, increasing the risk of the disease. Figure 2 displays the identified interactions (Figure 2A), corresponding odds ratio values (Figure 2B), and measures of interaction (Figure 2C). Interaction tables of 4 × 2, containing the number of subjects in each class, are provided in Appendix A.

Interactions with TC levels were observed for carriers of allele A (rs3825807), allele C (rs1994016), and allele T (rs7173743). For all these polymorphisms, the impact of the mentioned alleles on the increase in CAD risk was particularly evident in patients with TC levels ≥ 5 mmol/L, and the observed effect was consistently higher than what would be expected from the addition or multiplication of the effects of individual factors (Figure 2). Additionally, the carrier status of allele A of the rs3825807 polymorphism also showed synergy with elevated LDL cholesterol levels. In this case, as well, the values of interaction measures indicate synergy between both factors and the observed effect exceeds the sum/multiplication of the effects of individual factors (Figure 2).

### 2.4. Survival Analysis

The analysis did not reveal any significant association between investigated polymorphisms and the survival of patients over the 5- and 10-year periods of observation. Differences were not observed while comparing survival curves for all analyzed genotypes of a specific polymorphism (additive model) or when comparing heterozygotes plus the major allele homozygotes vs. minor allele homozygotes (recessive model). There was observed, however, a trend of improved survival in the 5-year follow-up for both the rs1994016 (TT genotype) and rs3825807 (GG genotype) polymorphisms. Interestingly, such a trend has not been found in the 10-year follow-up. The survival curves and results obtained from the χ^2^ and log-rank tests are presented in Figure 3, Figure 4 and Figure 5.

## 3. Discussion

Our study demonstrated an association between the carriage of the A allele of the rs3825807 *ADAMTS7* gene polymorphism and an increased risk of CAD. A similar relationship was also observed in the case of the carriage of the C allele of the rs1994016 SNP. The analysis of the rs7173743 polymorphism did not show any statistically significant association with CAD; however, we observed a tendency for the T allele carriers to occur more frequently in patients as compared to the healthy controls.

We also analyzed the potential impact of specified polymorphisms on the risk of CAD occurrence, considering previously approved and routinely used risk factors of this disease. The results showed the presence of a strong synergistic effect between the A allele carriage of the rs3825807 and elevated levels of total cholesterol and LDL cholesterol. In the case of the C allele of the rs1994016 SNP, we observed biological interactions with increased cholesterol, which potentially may predispose to CAD. Additionally, the T allele carriage of the rs7173743 polymorphism showed a synergistic effect on CAD with abnormal levels of cholesterol in serum samples.

Our study did not show any significant impact of *ADAMTS7* gene polymorphisms on the survival of patients with CAD. However, we observed a trend toward the better survival of TT homozygotes (rs1994016) and GG homozygotes (rs3825807) in the 5-year follow-up. Interestingly, in the 10-year follow-up, this trend has not been found, which may suggest that the genetic factor may have a greater significance on survival in the earlier stages of the disease.

Some studies from databases confirm, to some extent, the observations presented in our study. For instance, the rs3825807 variant (Ser214Pro) may influence the expression of *ADAMTS7* due to the substitution of a polar serine with a nonpolar proline (allele A to G) in the prodomain of the encoded protein [15]. The A allele is responsible for the increase in the expression of the gene and determines the maintenance of the catalytic activity of ADAMTS7. Moreover, the expression of this risk allele results in increased migration of smooth muscle cells of blood vessels [8]. Based on studies conducted on an animal model, it has been found that the carriage of the A allele of the rs3825807 *ADAMTS7* increases the risk of CAD and the development of atherosclerotic changes [15]. Similarly, in a group of non-smoking patients, the A allele carriage was associated with an increased risk of CAD [16]. In a cohort of Slovenian patients with type 2 diabetes, the AA genotype of rs3825807 was considered a genetic risk factor for myocardial infarction [17]. Additionally, studies in the Turkish population showed that AA homozygosity of the rs3825807 and CC of the rs1994016 in patients with peripheral artery disease significantly increases the level of *ADAMTS7* mRNA, leading to the disease development [12]. A similar effect was observed in the Chinese population in the case of the rs1994016 polymorphism, where the C allele was associated with an increase in ADAMTS7 concentration and the risk of acute coronary syndrome [18]. The strong pro-atherogenic effect of selected *ADAMTS7* polymorphisms may result from the promotion of processes underlying atherosclerotic plaque formation, such as thrombospondin-5 cleavage and migration of smooth muscle cells of blood vessels [9].

Referring to the presented biological interactions and synergistic effects of analyzed polymorphisms with lipid abnormalities, previous studies indicated that increased ADAMTS7 levels correlate positively with atherosclerotic plaque components that may determine instability, such as increased lipid concentration [19]. Nevertheless, understanding the pathomechanism of these associations requires further investigations. The above observations seem to be particularly important because understanding the functional significance of the discussed *ADAMTS7* variants may result in the development of new therapeutic methods in the future.

Previous studies in an animal model show that carrying the A allele of the rs3825807 *ADAMTS7* polymorphism increased not only the risk of CAD but also the risk of progression of atherosclerotic lesions, which undoubtedly would affect survival [15]. Indeed, a survival analysis of patients from the Caucasian population [5] showed that the AA genotype was an independent risk factor for death due to cardiovascular diseases in relation to the GG genotype (HR = 2.7, *p* = 0.025), with a follow-up period of 182 months (approx. 15 years). It has been speculated that the native, hyperfunctional A allele may accelerate vascular smooth muscular cell migration and lead to neointimal thickening, progression of atherosclerosis, and acute plaque events. Interestingly, our results showed a similar trend in the 5-year follow-up.

The limitation of our study is a relatively low number of study group participants, which may affect the statistical power of potential relationships between the analyzed SNPs and lipid metabolism parameters, the clinical phenotype of CAD, or the presented survival analysis. It is, however, worth emphasizing that our group was ethnically homogenous. In addition, the advantage of the study is the age of the patients (relatively young and comparable in the group) and that the clinical part of the study was conducted by professionals due to long-term collaboration with prestigious medical centers in Poland.

## 4. Materials and Methods

These case–control studies (association analysis) and prospective cohort studies (survival analysis) were conducted under STROBE guidelines. The case/cohort group included patients with CAD and healthy blood donors as a control group. Three SNPs of the ADAMTS7 gene were genotyped. The odds ratio values (OR with their 95% CI) were used to assess the risk of disease occurrence. Biological interactions between genetic and previously approved (routine) risk factors of CAD were evaluated by Rothman’s additive synergy index (SI), multiplicative synergy index (SIM), relative excess risk due to the interaction (RERI), and the proportion attributable to interaction (AP). Mortality of CAD was analyzed in a 5-year and 10-year observation, respectively.

### 4.1. Patients and Controls

The case–control study included 471 Caucasian subjects divided into two groups. The first group consisted of 231 patients with angiographically confirmed premature CAD, aged 44.37 ± 5.96 years and 55 years as the upper age limit of probands. The second group included 240 control blood donors (BD group), aged 43.35 ± 6.41 years, without symptoms of CAD, history of myocardial infarction (MI), stroke, and with no family history of cardiovascular diseases. Patients were recruited from the First Department and Clinic of Cardiology at the Upper Silesian Center of Cardiology in Katowice and the First Department of Cardiac Surgery at the Upper Silesian Center of Cardiology in Katowice. The control group was selected among blood donors of the Regional Centers of Blood Donation and Blood Treatment in Katowice and Raciborz, as previously described [20].

According to the guidelines of the Regional Centers of Blood Donation and Blood Treatment, the control group consisted of individuals without hypertension with systolic blood pressure < 140 and diastolic blood pressure < 90 on the day of blood collection, with other inclusion and exclusion criteria previously published [21].

The recruitment of the study groups was carried out from 2001 to 2011. The study was carried out in accordance with the Declaration of Helsinki and approved by the Bioethics Committee of the Medical University of Silesia in Katowice, Poland (Application KNW/0022/KB1/17/I/11). Prior to the study, written informed consent was obtained from all donors.

### 4.2. Serum Lipid Measurement

Analyses of lipid parameters included total cholesterol (TC), triglycerides (TG), and HDL cholesterol in blood serum by enzymatic colorimetric methods (Analco, Warsaw, Poland). The concentration of LDL cholesterol in serum was calculated using the Friedewald formula [22].

### 4.3. Genetic Analysis

Genomic DNA was extracted from peripheral leukocytes using the MasterPure genomic DNA purification kit (Epicentre Technologies, Madison, WI, USA). Three polymorphisms of the *ADAMTS7*, namely rs1994016, rs3825807, and rs7173743, were genotyped using the TaqMan^®^ Pre-designed SNP Genotyping Assay Kit (Applied Biosystems, Foster City, CA, USA). The 20-µL reaction mix consisted of 1 µL template DNA (15 ng/µL), 10 µL TaqMan^®^ Genotyping Master Mix (Cat. # 4371355), 1 µL probe (TaqMan^®^ Pre-designed SNP Genotyping Assay), and 8 µL deionized water. The probe was diluted (1:1) in TE buffer (10 mM Tris-HCl, pH 8.0, 0.1 mM EDTA) and used immediately for the reaction. PCR was performed according to the manufacturer’s specifications. Efficient genotyping was performed for about 94–97% of study participants, with the accuracy verified by repeated genotyping for 15% and the repeatability of the results for 100%.

### 4.4. Statistical Analysis

The data analysis was conducted using Statistica 13.0 software (STATSOFT, Tulsa, OK, USA). For quantitative data, the Shapiro–Wilk test was applied to check the normality of the distribution. The comparison of quantitative variables between CAD and control groups was performed using the Mann–Whitney U test (non-normal distribution) or Student’s *t*-test (normal distribution). Allele frequencies were inferred from the genotype distribution. Hardy–Weinberg equilibrium testing, comparisons of genotypes, and allele frequencies between cases and control subjects were calculated using an χ^2^ test in the case of univariate analysis. The results were considered statistically significant at *p* < 0.05. For biological interaction analyses, the Bonferroni correction was applied, and the results were considered statistically significant at *p* < 0.025. The odds ratios (OR with 95% confidence intervals, CI) were computed using univariate and multiple logistic regression analysis after adjustment for age, sex, and previously approved CAD risk factors. If the number of individuals in the analyzed subgroups was zero, the risk ratio values (RR with 95% CI) were used. Haplotype blocks were estimated using HaploView 4.2. [23] software. The algorithm of Gabriel et al. [24] was used for calculations of linkage disequilibrium (D′ and R^2^) and haplotype frequencies.

Association analysis was performed according to two different models, i.e., additive: dominant (major allele homozygotes vs. minor allele homozygotes plus the heterozygotes), and recessive model (heterozygotes plus the major allele homozygotes vs. minor allele homozygotes) [25].

The analysis of synergy in multiplicative and additive models was performed to interpret the amount of interaction, according to the recommendations of Knol et al. [26] and Knol and VanderWeele [27]. Asymmetric confidence intervals (CI) for additive interaction parameters (SI) were determined using the model of Zou [28]. Synergy indexes were calculated based on OR values from 4 × 2 tables using the following formulas:

-For SIM (multiplicative synergy index) [29]:


SIM = OR11/OR01 × OR10


-For SI (Rothman’s additive synergy index):


SI = OR11 − 1/(OR01 − 1) + (OR10 − 1)


-For RERI (relative excess risk due to interaction):


RERI = OR11 − OR10 − OR01 + 1


-For AP (proportion attributable to interaction):


AP = RERI/OR11


### 4.5. Survival Analysis

In the survival analysis, the endpoint was the death of the patient due to cardiovascular episodes, according to the International Statistical Classification of Diseases and Related Health Problems (ICD-10). Data on the date and causes of death of patients were obtained from the Katowice City Hall and the Central Statistical Office of Poland. Complete observation referred to cases where the endpoint occurred, while censored observation was defined as cases where the patient survived or died due to different reasons other than cardiovascular diseases.

The Kaplan–Meier estimator was used for survival analysis. Statistical differences between survival curves for individual genotypes (additive model) were calculated using the χ^2^ test, while differences in the recessive model (heterozygotes plus the major allele homozygotes vs. minor allele homozygotes) were assessed in the log-rank test. The results were considered statistically significant at *p* < 0.05. Analyses were conducted for both 5-year and 10-year survival periods. Survival was defined as the period (in yrs) from angiographic confirmation of CAD until death from cardiovascular episodes.

## 5. Conclusions

In conclusion, this study revealed that carrying the A allele of rs3825807 and the C allele of rs1994016 *ADAMTS7* may be a predisposition to CAD occurrence in the Polish population. Moreover, considering the contribution of previously approved and routine CAD risk factors, all three variants, i.e., rs3825807, rs1994016, and rs7173743, increased the risk of CAD in synergy with abnormalities of blood lipids concentration.

## Figures and Tables

**Figure 1 ijms-25-02274-f001:**
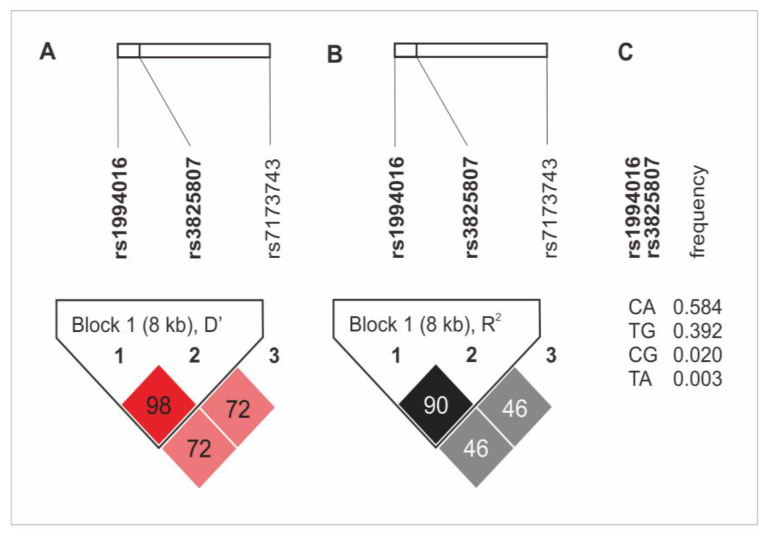
Haplotype analysis of *ADAMTS7* gene polymorphisms in the study group. (**A**) D’ values; (**B**) R^2^ values; (**C**) frequency of diplotypes of rs1994016 and rs3825807 SNPs.

**Figure 2 ijms-25-02274-f002:**
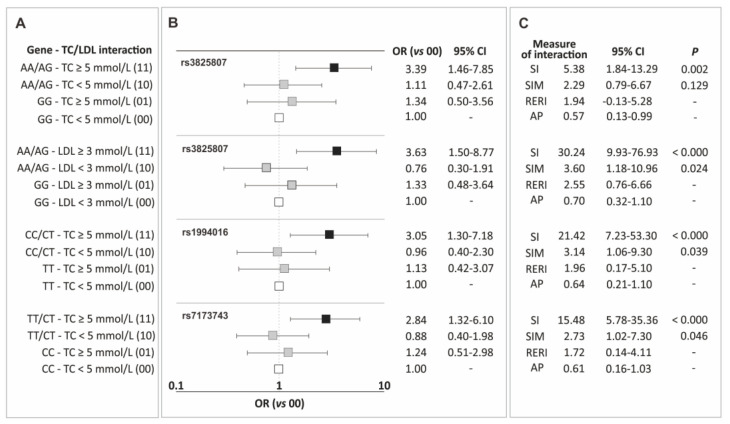
Identified interactions of the *ADAMTS7* gene polymorphisms and elevated levels of total cholesterol and LDL cholesterol (**A**), corresponding odds ratio values (**B**), and measures of interaction (**C**). Legend: AP, attributable proportion due to interaction; CI, confidence interval; LDL, low-density lipoprotein; OR, odds ratio; RERI, relative excess risk due to interaction; SI, synergy index; SIM, multiplicative synergy index; TC, total cholesterol.

**Figure 3 ijms-25-02274-f003:**
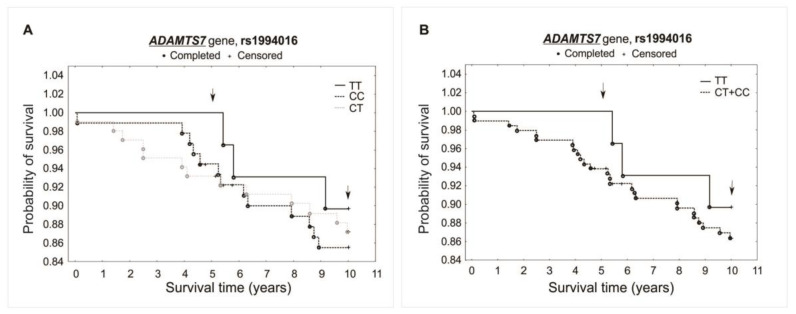
Kaplan–Meier survival curves for CAD patients by rs1994016: (**A**) additive model: for 5-year survival, χ^2^ = 2.094, *p* = 0.351; for 10-year survival, χ^2^ = 0.405, *p* = 0.817. (**B**) Recessive model: for 5-year survival, the log-rank statistic was −1.374, *p* = 0.170; for 10-year survival, the log-rank statistic was −0.523, *p* = 0.601.

**Figure 4 ijms-25-02274-f004:**
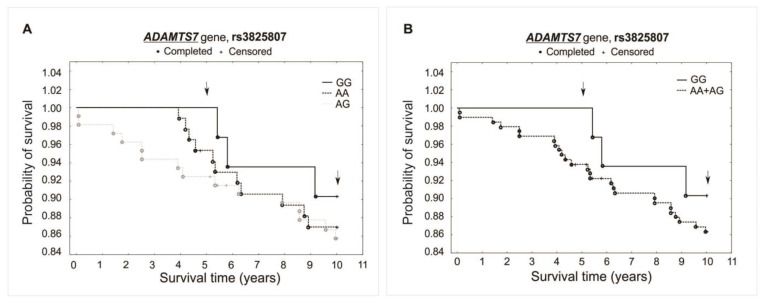
Kaplan–Meier survival curves for CAD patients by rs3825807: (**A**) additive model: for 5-year survival, χ^2^ = 2.928, *p* = 0.231; for 10-year survival, χ^2^ = 0.609, *p* = 0.737. (**B**) Recessive model: for 5-year survival, the log-rank statistic was −1.424, *p* = 0.155; for 10-year survival, the log-rank statistic was −0.650, *p* = 0.516.

**Figure 5 ijms-25-02274-f005:**
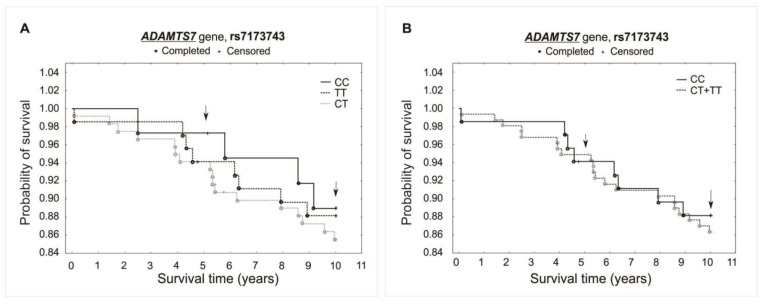
Kaplan–Meier survival curves for CAD patients by rs7173743: (**A**) additive model: for 5-year survival, χ^2^ = 0.604, *p* = 0.739; for 10-year survival, χ^2^ = 0.506, *p* = 0.776. (**B**) Recessive model: for 5-year survival, the log-rank statistic was −0.206, *p* = 0.837; for 10-year survival, the log-rank statistic was 0.359, *p* = 0.719.

**Table 1 ijms-25-02274-t001:** Genotype and allele frequencies of *ADAMTS7* in CAD patients and controls.

Genotype/Allele	CAD n (%)	Controls n (%)	Inheritance Model	OR (95% CI)	*p*
*rs1994016*					
CC	85 (47.93)	84 (36.21)	Dominant vs. CT+TT	1.14 (0.77–1.66)	0.517
CT	104 (39.17)	101 (43.53)	Additive vs. CC	1.02 (0.68–1.53)	0.933
TT	28 (12.90)	47 (20.26)	Recessive vs. CT+CC	0.58 (0.35–0.97)	0.036 *
CT +CC	189 (87.10)	185 (79.74)	Recessive vs. TT	1.72 (1.03–2.86)	0.036 *
C	274 (63.13)	269 (57.97)	-	1.24 (0.95–1.62)	0.114
T	160 (36.87)	195 (42.03)	-	0.81 (0.59–1.10)	0.114
*rs3825807*					
AA	80 (36.70)	78 (33.48)	Dominant vs. AG + GG	1.15 (0.78–1.70)	0.474
AG	108 (49.54)	106 (45.49)	Additive vs. AA	0.99 (0.66–1.50)	0.975
GG	30 (13.76)	49 (21.03)	Recessive vs. AA + AG	0.60 (0.36–0.99)	0.042 *
AA+ AG	188 (86.24)	184 (78.97)	Recessive vs. GG	1.67 (1.02–2.74)	0.042 *
A	268 (61.47)	262 (56.22)	-	1.24 (0.95–1.62)	0.110
G	168 (38.53)	204 (43.78)	-	0.81 (0.62–1.05)	0.110
*rs7173743*					
TT	65 (29.82)	75 (32.19)	Dominant vs. TC + CC	0.90 (0.60–1.34)	0.586
TC	117 (53.67)	103 (44.21)	Additive vs. TT	1.31 (0.86–2.05)	0.212
CC	36 (16.51)	55 (23.60)	Recessive vs. TT + TC	0.64 (0.40–1.02)	0.068
TT+ TC	182 (83.49)	178 (76.40)	Recessive vs. CC	1.59 (0.98–2.50)	0.068
T	247 (56.65)	253 (54.29)	-	1.1 (0.85–1.43)	0.476
C	189 (43.35)	213 (45.71)	-	0.91 (0.70–1.18)	0.476

*—differences statistically significant; OR—odds ratio; CAD—coronary artery disease (patient group).

## Data Availability

Dataset available on request from the authors.

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
