# Peer review of "The Association of ADAMTS7 Gene Polymorphisms with the Risk of Coronary Artery Disease Occurrence and Cardiovascular Survival in the Polish Population: A Case-Control and a Prospective Cohort Study"

_ijms, 2024, doi:10.3390/ijms25042274_

Round 1

Reviewer 1 Report

Comments and Suggestions for Authors

I only have two comments:

1.     Association of polymorphisms in ADAMTS-7 gene with the susceptibility to coronary artery disease and cardiovascular survival is already a well explored research area in literature. There does not seem to be any new novel findings that the study seems to add.

2.     Many findings can be represented better in figures instead of tables. Why do we need 7 tables? That is too much for the main text.

Author Response

Dear Reviewer,

Thank you very much for reading and reviewing our manuscript. This will help us to reach better scientific level. We revised manuscript and made some changes. We send revised version of the manuscript. All changes were marked in red.

  1. Association of polymorphisms in ADAMTS-7 gene with the susceptibility to coronary artery disease and cardiovascular survival is already a well explored research area in literature. There does not seem to be any new novel findings that the study seems to add.

 We decided to investigate the selected polymorphisms of the ADAMTS7 gene (rs1994016, rs3825807, and rs7173743) because they have not been previously studied in the Polish population in the context of CAD risk. The literature also indicates that only rs3825807 variant is associated with patient survival, the other polymorphisms have not yet been analysed. Moreover, this is the first paper describing possible biological interactions between genotypic variants of selected polymorphisms and traditional risk factors of CAD.

  1. Many findings can be represented better in figures instead of tables. Why do we need 7 tables? That is too much for the main text.

Raw data for biological interactions (tables 2-7) has been transferred to the Supplementary Material, and a new figure summarizing the results of the presented interactions has been added to the main text of our paper. Moreover, we have edited the description of the biological interaction results, trying to make them more clear.

Reviewer 2 Report

Comments and Suggestions for Authors

General comments:

The

 Association of ADAMTS7 gene polymorphisms with the risk of 2 CAD occurrence and cardiovascular survival in the Polish 3 population: a case-control and a prospective cohort study is an interesting as well as well- written RESEARCH paper. The primary objective of the study was to assess the aim of this study was to investigate 64 the impact of the rs1994016, rs3825807, and rs7173743 variants of the ADAMTS7 gene on 65 the risk of CAD in the case-control model.

Comment 1: Sample size is appropriate for the purpose of the study (231 cases, 240 controls).

Comment 2: Abstract section is rather well written. Methods, Results and Conclusion sections are clear.

Comment 3: Introduction section is rather well written regarding the structure, and the aim should be revised - the aim of the study should be defined more clear-cut (see also comment 2) -

Comment 4: Authors should add few details about atherosclerotic markers – I would suggest adding another paragraph (after first, before second paragraph). There are some review papers that cover that aspects:

Ref 1: Libby P. The changing Nature of atherosclerosis: what we thought we knew, what we think we know, and what we have to learn.  Eur Heart J. 2021 Dec 14;42(47):4781-4782.

Ref 2: Tibaut M, Caprnda M, Kubatka P, Sinkovič A, Valentova V, Filipova S, Gazdikova K, Gaspar L, Mozos I, Egom EE, Rodrigo L, Kruzliak P, Petrovič D. Markers of Atherosclerosis: Part 2-Genetic and Imaging Markers. Heart Lung Circ. 2018 Sep 29. pii: S1443-9506(18)31914-0. doi: 10.1016/j.hlc.2018.09.006

Ref 3: Stone PH, Libby P, Boden WE. Fundamental Pathobiology of Coronary Atherosclerosis and Clinical Implications for Chronic Ischemic Heart Disease Management-The Plaque Hypothesis: A Narrative Review. JAMA Cardiol. 2023 Feb 1;8(2):192-201. doi: 10.1001/jamacardio.2022.3926. PMID: 36515941 Review.

Comment 5: Material and Methods are rather well defined.

Comment 6: Results section: results are rather well presented. Regarding the sample size, power calculation need to be added

Comment 7: Discussion section: Rather well written

Comment 8: References are appropriate although few should be added (see comment 4)

Author Response

Dear Reviewer,

Thank you very much for reading and reviewing our manuscript. This will help us to reach better scientific level. We revised manuscript and made some changes. We send revised version of the manuscript. All changes were marked in red.

 Association of ADAMTS7 gene polymorphisms with the risk of 2 CAD occurrence and cardiovascular survival in the Polish 3 population: a case-control and a prospective cohort study is an interesting as well as well- written RESEARCH paper. The primary objective of the study was to assess the aim of this study was to investigate 64 the impact of the rs1994016, rs3825807, and rs7173743 variants of the ADAMTS7 gene on 65 the risk of CAD in the case-control model.

Comment 1: Sample size is appropriate for the purpose of the study (231 cases, 240 controls).

Comment 2: Abstract section is rather well written. Methods, Results and Conclusion sections are clear.

Comment 3: Introduction section is rather well written regarding the structure, and the aim should be revised - the aim of the study should be defined more clear-cut (see also comment 2) –

We made changes to the aim of our study, trying to make it more clear.

Comment 4: Authors should add few details about atherosclerotic markers – I would suggest adding another paragraph (after first, before second paragraph). There are some review papers that cover that aspects:

Ref 1: Libby P. The changing Nature of atherosclerosis: what we thought we knew, what we think we know, and what we have to learn.  Eur Heart J. 2021 Dec 14;42(47):4781-4782.

Ref 2: Tibaut M, Caprnda M, Kubatka P, Sinkovič A, Valentova V, Filipova S, Gazdikova K, Gaspar L, Mozos I, Egom EE, Rodrigo L, Kruzliak P, Petrovič D. Markers of Atherosclerosis: Part 2-Genetic and Imaging Markers. Heart Lung Circ. 2018 Sep 29. pii: S1443-9506(18)31914-0. doi: 10.1016/j.hlc.2018.09.006

Ref 3: Stone PH, Libby P, Boden WE. Fundamental Pathobiology of Coronary Atherosclerosis and Clinical Implications for Chronic Ischemic Heart Disease Management-The Plaque Hypothesis: A Narrative Review. JAMA Cardiol. 2023 Feb 1;8(2):192-201. doi: 10.1001/jamacardio.2022.3926. PMID: 36515941 Review.

Comment 5: Material and Methods are rather well defined.

As suggested by the Reviewer, we added another paragraph to the Introduction Section, and the above review papers to the bibliography.

Comment 6: Results section: results are rather well presented. Regarding the sample size, power calculation need to be added

We added information about power for our case-control study to the Results Section.

Comment 7: Discussion section: Rather well written

Comment 8: References are appropriate although few should be added (see comment 4)

We supplemented our references with the above review articles.

Round 2

Reviewer 1 Report

Comments and Suggestions for Authors

The manuscript can be accepted in present form

Reviewer 2 Report

Comments and Suggestions for Authors

Authors revised the paper well